# Electrochemical Evaluation of a Multi-Site Clinical Depth Recording Electrode for Monitoring Cerebral Tissue Oxygen

**DOI:** 10.3390/mi11070632

**Published:** 2020-06-28

**Authors:** Ana Ledo, Eliana Fernandes, Jorge E. Quintero, Greg A. Gerhardt, Rui M. Barbosa

**Affiliations:** 1Faculty of Pharmacy, University of Coimbra, 3000-548 Coimbra, Portugal; eliana.fernandes2604@outlook.com (E.F.); rbarbosa@ff.uc.pt (R.M.B.); 2Center for Neuroscience and Cell Biology, University of Coimbra, 3004-504 Coimbra, Portugal; 3Center for Innovative Biomedicine and Biotechnology, University of Coimbra, 3004-504 Coimbra, Portugal; 4Department of Neurosurgery, University of Kentucky Medical Center, Lexington, KY 40536, USA; george.quintero@uky.edu (J.E.Q.); gregg@uky.edu (G.A.G.); 5Department of Neuroscience, University of Kentucky Medical Center, Lexington, KY 40536, USA

**Keywords:** brain tissue oxygen, in vivo monitoring, multi-site clinical depth electrode

## Abstract

The intracranial measurement of local cerebral tissue oxygen levels—PbtO_2_—has become a useful tool for the critical care unit to investigate severe trauma and ischemia injury in patients. Our preliminary work in animal models supports the hypothesis that multi-site depth electrode recording of PbtO_2_ may give surgeons and critical care providers needed information about brain viability and the capacity for better recovery. Here, we present a surface morphology characterization and an electrochemical evaluation of the analytical properties toward oxygen detection of an FDA-approved, commercially available, clinical grade depth recording electrode comprising 12 Pt recording contacts. We found that the surface of the recording sites is composed of a thin film of smooth Pt and that the electrochemical behavior evaluated by cyclic voltammetry in acidic and neutral electrolyte is typical of polycrystalline Pt surface. The smoothness of the Pt surface was further corroborated by determination of the electrochemical active surface, confirming a roughness factor of 0.9. At an optimal working potential of −0.6 V vs. Ag/AgCl, the sensor displayed suitable values of sensitivity and limit of detection for in vivo PbtO_2_ measurements. Based on the reported catalytical properties of Pt toward the electroreduction reaction of O_2_, we propose that these probes could be repurposed for multisite monitoring of PbtO_2_ in vivo in the human brain.

## 1. Introduction

Monitoring local cerebral tissue oxygen levels—PbtO_2_—is increasingly used in neurological intensive care units to guide therapeutic strategies aimed at maintaining O_2_ levels above threshold, namely in patients suffering of severe acute brain conditions such as traumatic brain injury (TBI) and aneurysmal subarachnoid hemorrhage (aSAH), and during certain neurosurgery procedures [1,2,3]. The rationale for this comes from studies showing improved outcome in patients with brain O_2_ monitoring and PbtO_2_ targeted therapeutic approaches [4,5]. Monitoring of PbtO_2_ has allowed the determination of the normal range of PbtO_2_ in the healthy brain tissue to be 25–30 mmHg [6,7,8,9], while values below 15 mmHg are typically associated with hypoxia and ischemia [10,11].

Due to the dependency of neuronal activity over oxidative metabolism for energy supply, monitoring PbtO_2_ can be a surrogate signal of neurovascular response and metabolic activity, both of which can become severely compromised in situations of acute brain injury [12,13]. In addition, cerebral ischemia is linked to the onset of secondary brain injury by predisposing brain tissue to an energetic crisis as well as contributing to the initiation of spreading cortical depolarizations [14,15]. These are all-or-none tissue level events characterized as near-complete breakdown of neuronal membrane potential which, in injured tissue, can initiate cell death cascades [16]. Finally, besides being a key metabolite in energy metabolism, O_2_ is also a substrate for multiple enzymes within cells and can act as a signal for genetic adaptation to situations of hypoxia, regulating gene expression via hypoxia-inducible factor (HIF) dependent pathways [17].

Currently approved methods for clinical monitoring of PbtO_2_ include invasive techniques such as amperometric sensors and optical sensors, and non-invasive techniques such as those using near infrared spectroscopy (NIRS) [18]. NIRS does not directly measure O_2_, but rather the level of hemoglobin saturation within a given tissue volume [19]. Among the invasive techniques, the amperometric LICOX^®^ probe by Integra^®^ LifeScience is considered by some to be the gold standard for PbtO_2_ monitoring [20]. This is a Clark-type sensor comprising cathode and anode electrodes encased in an 80 µm-thick polyethylene membrane [21] across which tissue O_2_ diffuses to then be detected in an electrochemical reduction reaction, producing an analytical redox current signal. Optical probes such as Neurovent^®^-PTO by Raumedics^®^ are based on fluorescence quenching by O_2_ of a probe [22]. Each one of these approaches has advantages and limitations, as expected with any other sensing technology. The amperometric and optical probes for focal PbtO_2_ and NIRS probes allow real-time monitoring of brain oxygenation in patients with variable temporal and spatial resolution. Both focal probes are invasive and are single site recording devices, and thus positioning of the probe limits the information obtained. On the other hand, NIRS probes are non-invasive and can assess several regions simultaneously but can experience contamination of the signal due to extracerebral circulation. Finally, focal PbtO_2_ probes are considered the most effective bedside method for detection of cerebral ischemia, which is not true for NIRS probes due to lack of standardization between commercial devices and undetermined threshold for ischemia [23].

Platinum recording surfaces display excellent properties for both stimulation and recording electrodes [24]. Clinical grade intracranial recording electrodes used for invasive monitoring of brain tissue electrical activity are typically composed of Pt on the recording sites [25]. This includes strip and grid subdural multi-electrode devices as well as intracranial multi-site recording electrodes for depth recordings. Considering that Pt displays electrocatalytic behavior toward the electroreduction of O_2_ [26,27], we propose that clinical grade recording electrodes can be used for amperometric monitoring of PbtO_2_, an application that has not, to the best of our knowledge, been explored. 

In the current work, we have characterized the electrochemical properties of an Auragen^TM^ depth electrode (Integra^®^ LifeScience, Princeton, NJ, USA) approved for brain mapping comprised of 12 cylindrical Pt recording sites. Furthermore, we investigated the analytical performance properties toward the reduction of O_2_. This is a critical first step towards the scaling up of our previous work in rodent models aimed at establishing fast sampling amperometry coupled to multi-site electrodes as a tool for concurrent electrophysiology and electrochemistry in clinical settings such as the neurocritical care unit and neurosurgery.

## 2. Materials and Methods

Reagents and Solutions: All reagents used were analytical grade and obtained from Merck (Algés, Portugal). Unless otherwise stated, all in vitro electrode evaluations were performed in PBS Lite solution, 0.05 M, pH 7.4 with the following composition: 10 mM Na_2_HPO_4_, 40 mM NaH_2_PO_4_, and 100 mM NaCl. Saturated O_2_ solutions for electrode calibration were prepared by bubbling PBS with 100% O_2_ (Air Liquide, Algés, Portugal) for 20 min, resulting in an O_2_ solution of 1.3 mM concentration at 22 °C [28]. Removal of O_2_ from solutions was achieved by purging with N_2_ (Air Liquide) for at least 20 min.

Auragen^TM^ Depth Electrode: In the current study we used a clinical grade flexible Auragen^TM^ depth electrode (ref. AU12D5L25) comprising 12 cylindrical Pt recording contacts with 2.5 mm in length and 5 mm spacing between consecutive recording sites (Appendix A) and gold connector contacts. The geometrical area of each lead was calculated to be 0.094 cm^2^ based on the measured diameter of 1.2 mm. The probe was used with no surface modifications or treatments.

Scanning Electron Microscopy and Elemental Composition Analysis: High-resolution scanning electron microscopy (SEM) was performed using a field emission scanning electron microscope coupled with energy dispersive X-ray spectroscopy (EDS) (Zeiss Merlin coupled to a GEMINI II column). The elemental composition was obtained from backscattered electron detection using EDS at 10 keV (X-Max, Oxford Instruments, High Wycombe, UK). Conductive carbon adhesive tabs were used to ground the electrode surface and secure the sample onto the specimen holder.

Electrochemical Instrumentation: Electrochemical characterization was performed on a MultiPalmSens4 Potentiostat equipped with a MUX8-R2 Multiplexer (PalmSens BV, Houten, The Netherlands) and controlled by MultiTrace software (PalmSens BV, The Netherlands). We used a three-electrode electrochemical cell comprising the depth electrode as working electrode, Ag/AgCl in 3M NaCl as reference electrode (RE-5B, BAS Inc, West Lafayette, IN, USA) and a Pt wire as auxiliary electrode. 

Electrode Calibration: The depth electrodes were calibrated to assess analytical performance toward O_2_ response. Calibrations were performed in 0.05 M PBS Lite pH 7.4 (20 mL) at room temperature (22 °C) with continuous stirring at low speed (240 rpm). Oxygen was removed by purging the solution with N_2_ for a minimum period of 20 min, after which the needle was removed from the solution and kept above the surface to decrease O_2_ back-diffusion. Once a stable baseline was obtained, 8.25 µM aliquots of the O_2_ saturated solution were added in 7 consecutive repetitions (concentration range 0–57.75 µM). The mean recording display frequency was set at 4 Hz.

Data Analyses: Data analyses were performed using MultiTrace (PalmSens BV, Houten, The Netherlands), OriginPro 2016 (OriginLab, Northampton, MA, USA) and GraphPad 5.0 (GraphPad Software, San Diego, CA, USA). Values are given as the mean ± SD. The number of repetitions is indicated in each individual determination. The sensitivity of depth electrode sites towards O_2_ reduction was determined by linear regression analysis in the range 0–60 μM. The limit of detection (LOD) was defined as the concentration that corresponds to a signal-to-noise ratio of 3, calculated using the expression:LOD = 3 × SD/m,
where SD is the standard deviation of the baseline (20 s interval) and m is the slope of the calibration curve obtained [29].

## 3. Results

### 3.1. Characterization of the Electrode Surface—Morphology and Chemical Analysis

To evaluate the morphology of the Pt surface of the depth electrode, we obtained SEM micrographs of recording sites. As shown in Figure 1A and B, the Pt surface or the recording site appears to be smooth. Higher amplification revealed that the Pt coverage is not completely uniform throughout the surface (Figure 1C). The elemental composition of the surface of the recording sites was analyzed by energy dispersive X-ray spectroscopy (EDS). As shown in Figure 1D, the active surface is primarily composed of Pt (approx. 80%), although C, O, and Al were also found to be present in lower proportions (approx. 9, 7, and 5%, respectively). To further investigate the composition of the surface, we determined the elemental composition of different regions of the surface, as indicated in Figure 1E. The lighter regions of the SEM (labeled “Spectrum 10” in Figure 1E) are composed primarily of Pt (approx. 93%) with some C contamination (Figure 1F), while the darker regions (labeled “Spectrum 9” in Figure 1E) are composed of Al and O (50 and 44%, respectively, Figure 1G). The pseudo-color map of the relative distribution of different elements on the EDS-analyzed surface (Figure 2) revealed that C is uniformly distributed over the surface and is likely a contaminant. Furthermore, Pt distribution is complimentary of that of Al and O, which overlap, suggesting that the Pt has been deposited over an aluminum oxide surface (most likely Al_2_O_3_), although small regions are not coated with Pt. Comparison of the surface morphology and elemental composition before and after electrochemical evaluation revealed no significant differences, indicating a stable Pt surface (Appendix A).

### 3.2. Electrochemical Active Surface Area

A standard electrochemical redox couple was used to determine the electrochemical behavior of the Pt surface of the recording sites of the depth electrode. Cyclic voltammetry was carried out in 5.0 mM hexaamineruthenium (III) chloride (Ru(III)(NH_3_)_6_) in 0.5 M KCl solution at scan rates from 25 to 200 mV s^−1^. As shown in Figure 2, the cyclic voltammograms revealed a conventional cyclic voltammetry behavior with well-defined symmetrical oxidation and reduction peaks appearing at 25 mV s^−1^. In addition, both the anodic and cathodic peak currents (*I*_p,a_ and *I*_p,c_, respectively) varied linearly with the square root of the scan rate (Figure 3 inset; R^2^ values of 0.999 for both *I*_p,a_ and *I*_p,c_) indicating that the process was diffusion-controlled. The average *I*_p,a_/*I*_p,c_ ratio was 0.8 ± 0.2 (*n* = 12), which is close to the theoretical value of 1 for a totally reversible reaction [30]. The E_1/2_ and E_pa_ – E_pc_ values were determined to be 182 ± 3 mV and 72 ± 2 mV (*n* = 12), respectively. 

The electrochemically active surface area of the Pt recording sites was estimated using the Randles–Sevick equation for a reversible oxidation-reduction reaction considering a diffusion coefficient of *D* = 7.1 × 10^−6^ cm^2^ s^−1^ [31]. The calculated surface area was 8.5 × 10^−2^ ± 1.0 × 10^−2^ cm^2^ corresponding to a surface roughness of 0.90 ± 0.1 (*n* = 12). This is in line with the smooth Pt surface observed in the SEM micrographs, as well as the Pt coverage determined from the EDS elemental analysis, which showed that the Pt coverage of the recording site is roughly 90% of the analyzed area, with Al and O making up most of the remaining area.

### 3.3. Electrochemical Behavior in Acidic Electrolyte and in Neutral PBS

The well-known characteristic cyclic voltammogram of Pt in acid solution was used to further examine the electrochemical behavior of the Pt recording site of the depth electrode. For this purpose, the probe was characterized by cyclic voltammetry in N_2_-purged H_2_SO_4_ (0.1 M). Figure 4A shows cyclic voltammograms recorded between −0.4 and 1.4 V *vs* Ag/AgCl at increasing scan rates (50–1000 mV s^−1^) of a single recording site. The typical cyclic voltammogram exhibited redox peaks at −0.08 and −0.2 V. Furthermore, the presence of the three distinct peaks for H^+^ desorption was clearly observed. An oxidation wave was observed for *E* > 0.5 V due to the formation of Pt oxide species Pt-O and Pt-OH, and there is a strong reduction peak at about 0.52 V corresponding to oxide reduction. 

We further characterized the electrode behavior in a neutral physiological-like media (0.05 M PBS Lite at pH 7.4), which simulates brain extracellular fluid. As shown in Figure 4B, increasing the electrolyte pH resulted in the expected negative shift in hydrogen adsorption/desorption and Pt-O formation/reduction peaks as well as a decrease in peak current width. In both electrolytes, the potential window—that is, the potential range between molecular hydrogen evolution and the evolution of molecular oxygen—is approximately 1.5 V.

### 3.4. Electrochemical Impedance Spectroscopy

Electrochemical impedance spectroscopy (EIS) allows the study of the physical and interfacial properties of electrochemical systems. Spectra were recorded in a N_2_-purged solution containing 5mM K_4_[Fe(II)(CN)_6_] and 5mM K_3_[Fe(III)(CN)_6_] in KCl 0.5M at room temperature by applying a sinusoidal wave of amplitude 10 mV between 100 kHz and 0.1 Hz (10 frequencies per decade) at the OCP (+0.24 V vs. Ag/AgCl). Before recording each spectrum, the electrode was held at this applied potential for 5 minutes.

The Bode plot (Figure 5A) and complex plane plot (Figure 5B) display the expected profile for a single step charge transfer process with diffusion of the reactants to the electrode surface. The complex plane plot shows the typical capacitive arc at high frequencies followed by a straight line (45°) at lower frequencies. The data were fitted to the Randles circuit [32] shown in the inset of Figure 5B, and consisting of the cell resistance (R_1_) in series with a combination of a constant phase element (Q_1_) in parallel with the series combination of a charge transfer resistance (*R*_2_) and a Warburg impedance element (*W*). The latter accounts for mass transfer limitations imposed by diffusion, which appear at lower frequencies. The values for the charge transfer resistance, Warburg coefficient and double layer capacitance from fitting to the equivalent electrical circuit are presented in Table 1.

The value of *Z*ʹ at 1 kHz is typically reported for impedance measurements on electrodes. In this work, the recording sites of the depth electrode showed a *Z*ʹ value of 5.4 ± 1.3 Ω·cm^2^ (77.4 ± 20.6 Ω before area normalization) at 1 kHz.

### 3.5. Oxygen Reduction Reaction at the Platinum Surface

To determine the most suitable working potential for monitoring O_2_ in vivo, we performed calibrations of the depth electrode at applied potentials ranging from 0.0 to −0.8 V vs. Ag/AgCl. The slopes corrected for electrochemical surface area are plotted in Figure 6A, revealing an increase in sensitivity as the applied potential decreased from 0.0 to −0.8 V. Although it seems tempting to use the highest value of −0.8 V, as seen in Figure 6B, the increase in sensitivity is accompanied by an increase in the baseline current. Considering the expected low values of brain tissue O_2_ and variations in the µM range, to optimize resolution and LOD, we chose −0.6 V vs. Ag/AgCl as the optimal applied potential. In Figure 6C, we show a representative calibration recording and respective calibration curve (inset). We observed linearity for the concentration range 0–50 µM and mean sensitivity of −1.2 ± 0.2 A M^−1^·cm^−2^ and mean LOD of 0.4 ± 0.1 µM (*n* = 4).

## 4. Discussion

Platinum surfaces are of paramount importance in applications such as those involving neural interfacing and design of electrochemical sensors and biosensors [24]. In the present study, we have investigated the electrochemical properties of a clinical depth recording electrode, in particular its suitability for monitoring *p*O_2_ in brain tissue. 

It is commonly accepted that PbtO_2_ reflects changes in metabolism and cerebrovascular response and monitoring this parameter has become increasingly standardized in neurocritical care [33]. Currently available probes for monitoring PbtO_2_ in situ in patients include the Clark-type amperometric Licox^®^ probe and the Neurovent^®^-PTO optical probe, both of which are single site recording devices [21,22]. Based on our previous experience with multisite Pt microelectrode arrays, we propose that clinical multisite recording electrodes such as the depth probe used here may be suitable for monitoring PbtO_2_ from multiple sites in the brain, offering a more integrated vision of brain oxygenation. 

The morphological evaluation of the recording surface of the depth electrode revealed a smooth surface of Pt over what appears to be an Al-oxide substrate, most likely Al_2_O_3_. The Pt coverage was shown not to be complete, accounting for roughly 90% of the surface area. This was further corroborated by determination of the electroactive surface area, which indicated a roughness factor of 0.9.

The electrochemical characterization of the Pt surface performed by cyclic voltammetry in acidic media revealed redox peaks at −0.08 and −0.2 V, which could be attributed to strong and weak proton adsorption on Pt surfaces with (100) and (110) basal planes of a polycrystalline structure, respectively [34,35]. Furthermore, the presence of the three distinct peaks for H^+^ desorption indicates a high-quality Pt surface [35,36]. An oxidation wave is observed for *E* > 0.5 V due to the formation of Pt oxide species Pt-O and Pt-OH, and there is a strong reduction peak at about 0.52 V corresponding to oxide reduction. Furthermore, the potential window of 1.5V did not widen in buffered aqueous solution corroborating the smooth structure of the Pt surface as nanostructure surfaces tend to show an increase in the potential width in neutral vs. acid media [35].

The electrochemical impedance spectroscopy revealed that the value of *Z*ʹ was 77.4 ± 20.6 Ω at 1 kHz, a value that is slightly lower than that reported for a Pt/Ir depth electrode with similar recording surface area [37], and much lower than that reported in our previous study (0.2 MΩ) for a smooth thin-film Pt microelectrode array [26]. As impedance is inversely proportional to recording site size but should be low as to decrease noise in recording [25,38]. In microelectrodes, where high impedance due to size can become a limiting factor, surface area is typically increased through roughening or functionalization [39,40].

The evaluation of the analytical performance toward the oxygen reduction reaction revealed that the optimal working potential for monitoring O_2_ is −0.6V vs. Ag/AgCl, in line with our previous observations. However, we found that the sensitivity was slightly lower than our previously reported value for thin-film Pt microelectrode arrays (3.2 ± 0.5 A M^−1^ cm^−2^), while the LOD is in a similar range as previously reported value of 0.3 ± 0.2 µM [26]. This supports the suitability of the present probe to monitor changes in PbtO_2_ in vivo in brain tissue, where basal values have been found to be approx. 30 µM [41]. We have previously shown that Pt electrode surfaces are optimal for the electrochemical reduction of O_2_, allowing direct and real-time monitoring of PbtO_2_ in brain tissue in rodent models [26]. Further validation of PbtO_2_ monitoring in using this depth recording electrode will be carried out in a swine model.

Considering that brain cerebrovascular function and neurometabolism are coupled to neuronal activity, there is great interest in simultaneous monitoring of electrical activity and PbtO_2_, as changes in PbtO_2_ result from both cerebral blood flow and oxidative metabolism [42,43,44]. Currently available invasive technology requires that two separate probes be implanted in distinct brain regions [45], which greatly hinders the establishment of correlations between changes in tissue oxygenation and neural activity. In line with previous observations [46,47,48], we demonstrated that high frequency amperometry can concurrently report PbtO_2_ and local field potential related currents in vivo in a rodent model of seizures [49]. Combined with other studies using enzyme-based biosensors and the same methodological approach, strong evidence supports that the high frequency component (>1 Hz) of an amperometric signal is a surrogate signal for local field potential [47,48,50]. This is an attractive approach toward simultaneous monitoring of electrical and chemical activity in vivo in brain tissue using a seamless methodology (amperometry) and a single probe that may have multiple recording sites [51].

## 5. Conclusions

We present an electrochemical evaluation of the analytical properties toward oxygen detection of a clinical grade depth recording electrode comprising 12 Pt recording contacts. Based on the reported catalytical properties of Pt toward the electroreduction reaction of O_2_, we propose that these probes could be repurposed for multisite monitoring of PbtO_2_ in vivo in the human brain. We found that the surface of the recording sites is composed of a thin film of smooth Pt and that the electrochemical behavior evaluated by cyclic voltammetry in acidic and neutral electrolytes is typical of a polycrystalline Pt surface. The smoothness of the Pt surface was further corroborated by determination of the electrochemical active surface, confirming a roughness factor of 0.9. At an optimal working potential of −0.6 V vs. Ag/AgCl, the sensor displayed suitable values of sensitivity and LOD, supporting its capability for monitoring PbtO_2_ in vivo in the brain. The repurposing of these probes from electrophysiology to electrochemical detection of O_2_ will allow seamless multisite monitoring of PbtO_2_ in clinical setting, which holds promise in the context of multimodal monitoring in neurocritical care where PbtO_2_ has become increasingly standardized following evidence of improved patient outcomes when PbtO_2_ targeted therapeutic approaches are used.

## Figures and Tables

**Figure 1 micromachines-11-00632-f001:**
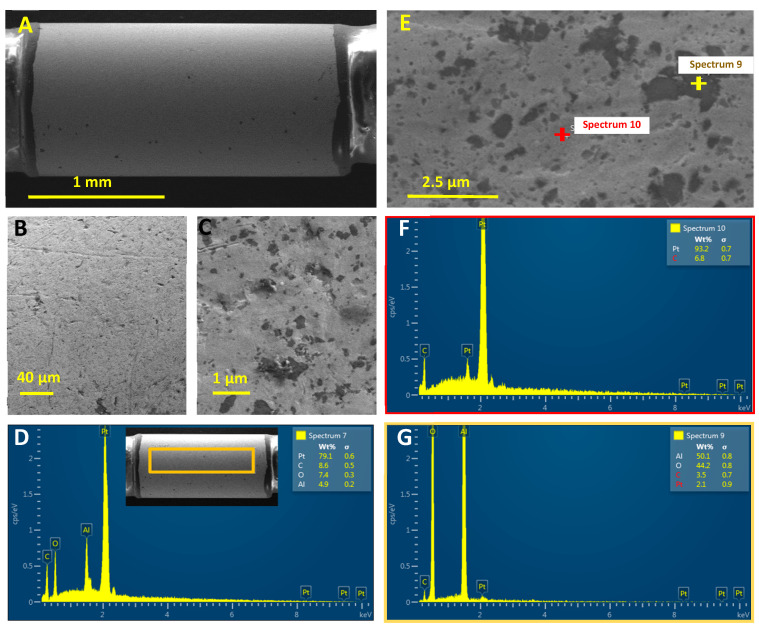
(**A**) General view of a recording site on the Auragen^TM^ Probe; (**B**) and (**C**) High-resolution micrographs of the recording surface; (**D**) EDS elemental analysis of the surface (ROI indicated in inset with orange rectangle); (**E**) Different regions of the recording surface were targeted for EDS elemental analysis, indicated with red and yellow “+” signs, respectively; (**F**) and (**G**) show the elemental composition spectrum of the lighter (conductive) regions (predominantly Pt) and of the darker (non-conductive) region (predominantly Al and O), respectively.

**Figure 2 micromachines-11-00632-f002:**
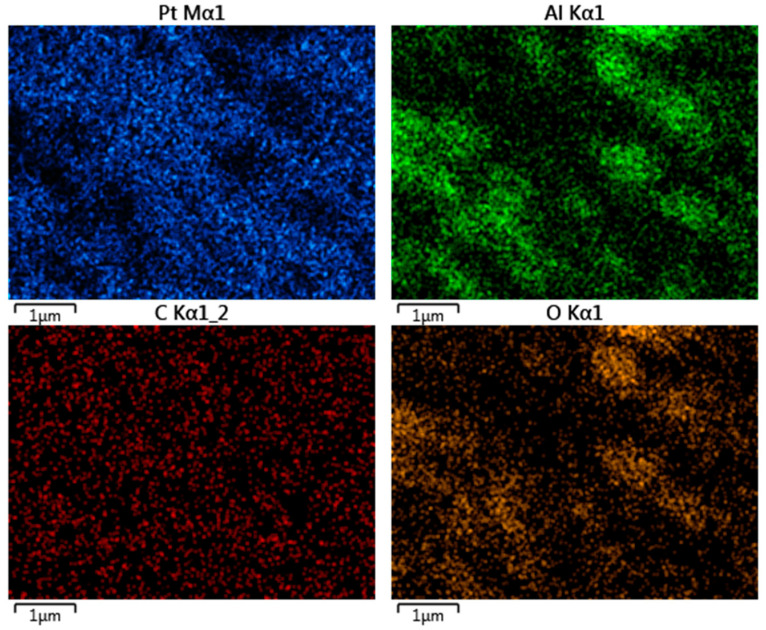
Pseudo-color map of the relative distribution of different elements on the EDS analyzed surface. Blue—platinum; green—aluminum; red—carbon, yellow—oxygen. Pt and Al distribution are complementary. Note that Al and O overlap, suggesting Al-O as a substrate for the Pt overcoat. Uniform distribution of C is in line with contamination.

**Figure 3 micromachines-11-00632-f003:**
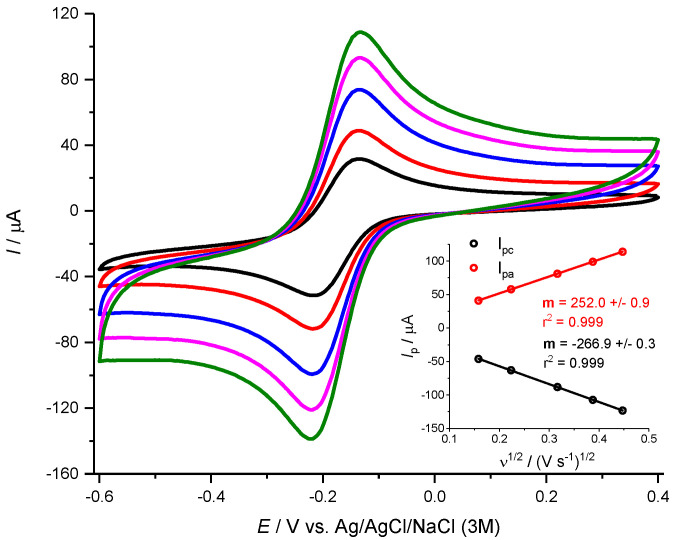
Reversible redox reaction of Ru(III)(NH_3_)_6_ in 0.5 M KCl at increasing scan rates (from 20 mV s^−1^ in black to 200 mV s^−1^ in green) and respective I*p* vs. ν^1/2^ plot for determination of the electrochemical surface area of the depth electrode recording site.

**Figure 4 micromachines-11-00632-f004:**
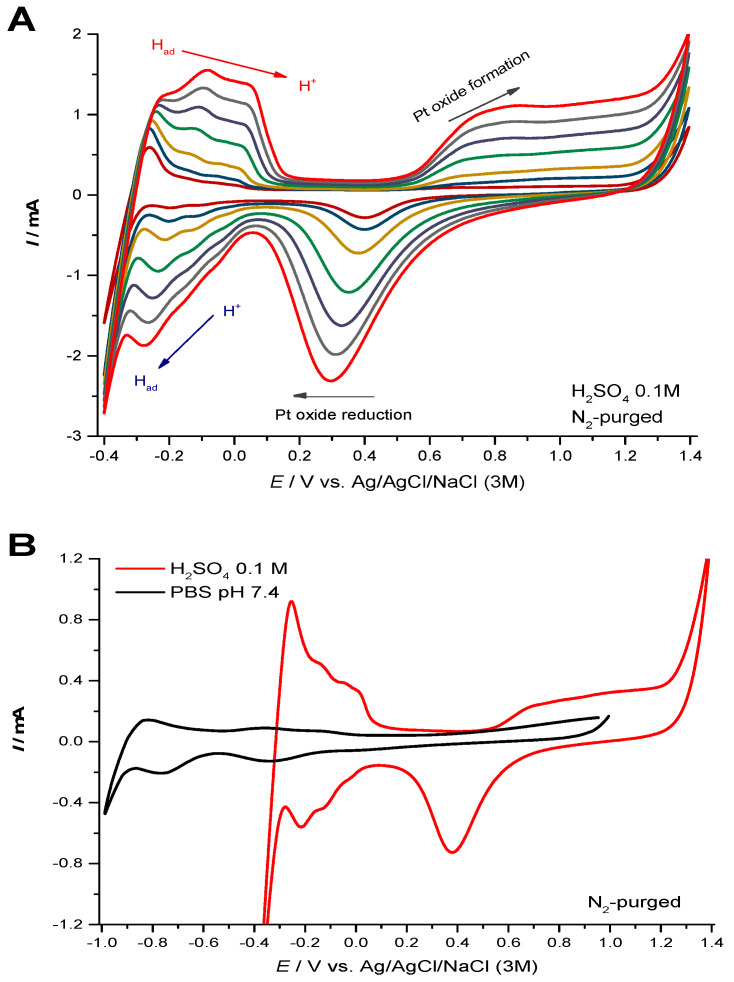
Electrochemical behavior in acidic and neutral electrolyte media. (**A**) Successive cyclic voltammograms (25^th^ scan) at increasing scan rates (50−1000 mV s^−1^) obtained in N_2_ saturated 0.1 M H_2_SO_4_, detailing the typical Pt oxide formation and reduction, proton adsorption (2 peaks) and desorption (3 peaks), and double layer zones. (**B**) Comparative CV plots (0.2 V s^−1^) recorded in N_2_-saturated 0.05 M, pH 7.4 PBS (black line) and N_2_-saturated 0.1 M, H_2_SO_4_ (red line) highlighting the positive shift in hydrogen evolution potential and increasing currents for Pt-oxide formation and reduction at lower pH on the Pt surface of the Integra Probe recording site.

**Figure 5 micromachines-11-00632-f005:**
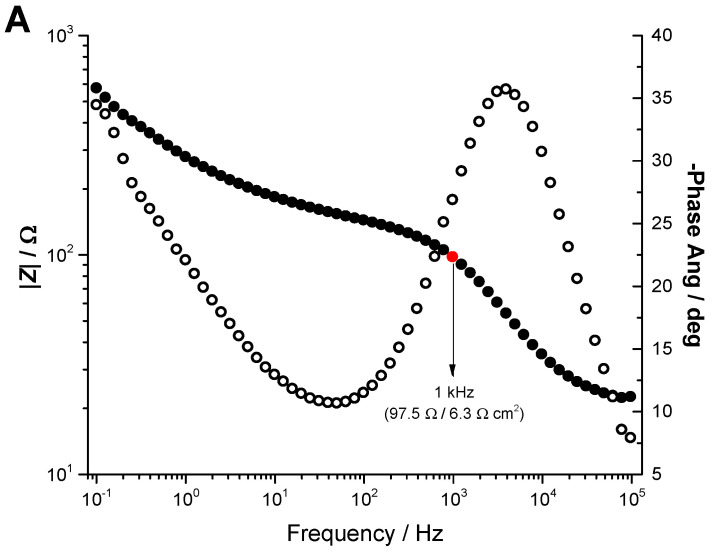
Electrochemical impedance spectroscopy measurements obtained in K_4_[Fe(II) (CN)_6_]/K_3_[Fe(III)(CN)_6_] (5 mM) in KCl 0.5 M. (**A**) Impedance−frequency plot (Bode plot). Filled circles represent |*Z*| values, and open circles are those obtained for the phase shift. The red circle highlights the |*Z*| value at 1 kHz. (**B**) Complex plane electrochemical impedance spectrum (Nyquist plot) of experimental data (open circles). Red line shows fitting to the electrical equivalent circuit shown in the inset. R1, solution resistance; R2, electron or charge transfer resistance; W, Warburg impedance element; Q, constant phase element.

**Figure 6 micromachines-11-00632-f006:**
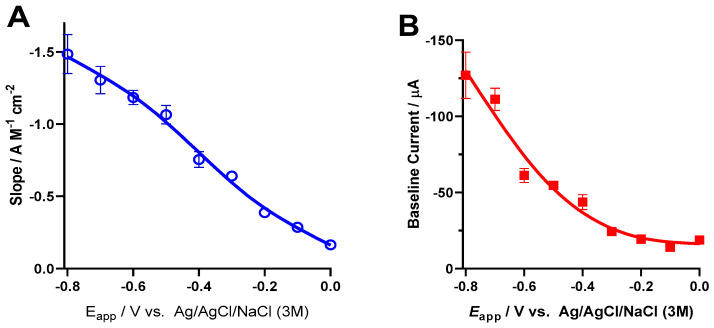
Electrochemical behavior of O_2_ reduction at the depth probe surface. (**A**) Average sensitivity as a function of the applied potential, obtained from calibration of 4 sites in PBS. (**B**) Average baseline current as a function of the applied potential, obtained in N_2_-purged PBS. (**C**) Representative calibration obtained at −0.6 V vs. Ag/AgCl and the calibration curve (inset) of a single recording site of the Integra probe. *n* = 4.

**Table 1 micromachines-11-00632-t001:** Summary of fitted parameter results for impedance spectroscopy measurements (*N* = 10) ^a^.

R1 (Ω)	R2	Q	n	AW (Ω·s^−0.5^)	Z at 1 kHz
18.98 ± 2.06	133.5 ± 33.4 Ω	47.39 ± 37.99 µF s^n−1^	0.68 ± 0.05	257 ± 14	77.44 ± 20.55 Ω
	9.3 ± 2.2 * Ω cm^2^	0.65 ± 0.49 * mF s^n−1^ cm^−2^			5.4 ± 1.3 * Ω cm^2^

^a^ Values shown with * are normalized by calculated surface area.

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
