# Peer review of "Electrochemical Evaluation of a Multi-Site Clinical Depth Recording Electrode for Monitoring Cerebral Tissue Oxygen"

_micromachines, 2020, doi:10.3390/mi11070632_

Round 1

Reviewer 1 Report

Major comment:

The use of PBS as a neutral physiological-like media to simulate brain extracellular fluid is not appropriate. As physiological as PBS may be commonly considered, it does not resemble the cerebrospinal fluid, which contains many more ions and molecules (e.g., dextrose) than PBS. In order to validate the measurements already performed in PBS, you should repeat them using artificial cerebrospinal fluid (ASCF) of standard composition.

Minor comments:

Page 1, lines 30-33: "Monitoring local cerebral tissue oxygen levels – PbtO2 – is increasingly used in neurological intensive care units to guide therapeutic strategies aimed at maintaining O2 levels above threshold, namely in patients suffering sever acute brain disease (traumatic brain injury (TBI) and aneurysmal subarachnoid hemorrhage (aSAH), for example) and during certain neurosurgery procedures [1–3]."

I suggest you rephrase to read:

"Monitoring of local cerebral tissue oxygen levels – PbtO2 – is increasingly used in neurological intensive care units to guide therapeutic strategies aimed at maintaining O2 levels above threshold in patients suffering sever acute brain conditions such as traumatic brain injury (TBI) and aneurysmal 32 subarachnoid hemorrhage (aSAH), and during certain neurosurgery procedures [1–3].

Page 2, line 72: Please, use the full spell for electrophysiology and electrochemistry, rather than e-physiology and e-chemistry, which sound colloquial.

Author Response

Reviewer 1)

Major comment:

The use of PBS as a neutral physiological-like media to simulate brain extracellular fluid is not appropriate. As physiological as PBS may be commonly considered, it does not resemble the cerebrospinal fluid, which contains many more ions and molecules (e.g., dextrose) than PBS. In order to validate the measurements already performed in PBS, you should repeat them using artificial cerebrospinal fluid (ASCF) of standard composition.

R: Although aCSF is without a doubt the most similar to brain interstitial fluid, PBS is commonly used in the analytical evaluation of amperometric sensors and biosensors. The major issue with aCSF in this particular case is pH buffering which is achieved through continuous bubbling of carbox or other gas mixture containing 5% CO2. In the case of O2 sensors and their analytical evaluation/response to O2, continuous bubbling with carbox is not a viable option. Adding organic buffers such as HEPES is not an option due to the fact that it is electroactive. We would like to highlight that PBS has been widely used as a standard media for calibration of amperometric (bio)sensors for nitric oxide and oxygen, as well as for hydrogen peroxide, ascorbate, glutamate, glucose and lactate, to site only a few.

References:

[1]          N.R. Ferreira, A. Ledo, J.G. Frade, G.A. Gerhardt, J. Laranjinha, R.M. Barbosa, Electrochemical measurement of endogenously produced nitric oxide in brain slices using Nafion/o-phenylenediamine modified carbon fiber microelectrodes, Anal. Chim. Acta. 535 (2005) 1–7. https://doi.org/10.1016/j.aca.2004.12.017.

[2]          A. Ledo, R.M. Barbosa, G.A. Gerhardt, E. Cadenas, J. Laranjinha, Concentration dynamics of nitric oxide in rat hippocampal subregions evoked by stimulation of the NMDA glutamate receptor, Proc. Natl. Acad. Sci. 102 (2005) 17483–17488. https://doi.org/10.1073/pnas.0503624102.

[3]          A. Ledo, R. Barbosa, E. Cadenas, J. Laranjinha, Dynamic and interacting profiles of .NO and O2 in rat hippocampal slices, Free Radic. Biol. Med. 48 (2010) 1044–1050. https://doi.org/10.1016/j.freeradbiomed.2010.01.024.

[4]          C.F. Lourenço, A. Ledo, J. Laranjinha, G.A. Gerhardt, R.M. Barbosa, Microelectrode array biosensor for high-resolution measurements of extracellular glucose in the brain, Sensors Actuators, B Chem. 237 (2016) 298–307. https://doi.org/10.1016/j.snb.2016.06.083.

[5]          A. Ledo, C.F. Lourenço, J. Laranjinha, C.M.A. Brett, G.A. Gerhardt, R.M. Barbosa, Ceramic-Based Multisite Platinum Microelectrode Arrays: Morphological Characteristics and Electrochemical Performance for Extracellular Oxygen Measurements in Brain Tissue, Anal. Chem. 89 (2017) 1674–1683. https://doi.org/10.1021/acs.analchem.6b03772.

[6]          N.R. Ferreira, A. Ledo, J. Laranjinha, G.A. Gerhardt, R.M. Barbosa, Simultaneous measurements of ascorbate and glutamate in vivo in the rat brain using carbon fiber nanocomposite sensors and microbiosensor arrays, Bioelectrochemistry. 121 (2018) 142–150. https://doi.org/10.1016/j.bioelechem.2018.01.009.

[7]          C.F. Lourenço, M. Caetano, A. Ledo, R.M. Barbosa, Platinized carbon fiber-based glucose microbiosensor designed for metabolic studies in brain slices, Bioelectrochemistry. 130 (2019) 107325. https://doi.org/10.1016/j.bioelechem.2019.06.010.

[8]          A. Ledo, E. Fernandes, C.M.A. Brett, R.M. Barbosa, Enhanced Selectivity and Stability of Ruthenium Purple-Modified Carbon Fiber Microelectrodes for Detection of Hydrogen Peroxide in Brain Tissue, Sensors Actuators B. Chem. (2020) 127899. https://doi.org/10.1016/j.snb.2020.127899.

[9]          C.F. Lourenço, A. Ledo, G.A. Gerhardt, J. Laranjinha, R.M. Barbosa, Neurometabolic and electrophysiological changes during cortical spreading depolarization: Multimodal approach based on a lactate-glucose dual microbiosensor arrays, Sci. Rep. 7 (2017). https://doi.org/10.1038/s41598-017-07119-6.

Minor comments:

Page 1, lines 30-33: "Monitoring local cerebral tissue oxygen levels – PbtO2 – is increasingly used in neurological intensive care units to guide therapeutic strategies aimed at maintaining O2 levels above threshold, namely in patients suffering sever acute brain disease (traumatic brain injury (TBI) and aneurysmal subarachnoid hemorrhage (aSAH), for example) and during certain neurosurgery procedures [1–3]."

I suggest you rephrase to read:

"Monitoring of local cerebral tissue oxygen levels – PbtO2 – is increasingly used in neurological intensive care units to guide therapeutic strategies aimed at maintaining O2 levels above threshold in patients suffering sever acute brain conditions such as traumatic brain injury (TBI) and aneurysmal 32 subarachnoid hemorrhage (aSAH), and during certain neurosurgery procedures [1–3].

R: We have changed the text accordingly.

Page 2, line 72: Please, use the full spell for electrophysiology and electrochemistry, rather than e-physiology and e-chemistry, which sound colloquial.

R: We have changed the text accordingly.

Reviewer 2 Report

Comments:

Title of the paper:
Electrochemical Evaluation of a Clinical Depth Recording Electrode for Monitoring Cerebral Tissue Oxygen

2-4 The title itself is not reflecting that the paper describes a multi site recording probe (electrode), also the keywords may be improved in this respect

Other detailed remarks and questions for the authors

Introduction:

48 Methods approved for clinical monitoring of PbtO2?
57-58 This is obvious and needs to be improved with some more consistent content related to the reference (23)

Material and Methods

81-85 Description of the depth electrode is very limited, a lot of suppositions remains to the reader

Results

128-129 The Pt and other layers are fabricated by the manufacturer and should be part of the Material and Methods; at least the thickness for the Pt and insulating layers must be known and then investigated in the Morphology and Chemical Analysis section.
129 What is the meaning of « albite unevenly »?

Author Response

Reviewer 2)

Title of the paper:
Electrochemical Evaluation of a Clinical Depth Recording Electrode for Monitoring Cerebral Tissue Oxygen

2-4 The title itself is not reflecting that the paper describes a multi site recording probe (electrode), also the keywords may be improved in this respect

R: We have changed the title and keywords to better reflect the multi-site nature of the recording probe.

Other detailed remarks and questions for the authors

Introduction:

48 Methods approved for clinical monitoring of PbtO2?

R: yes, these are methods that are clinically approved and current in use for monitoring PbtO2. We have clarified this in the text.

57-58 This is obvious and needs to be improved with some more consistent content related to the reference (23)

R: We have added information regarding limitations and advantages of the methods.

“. The amperometric and optical probes for focal PbtO2 and NIRS probes allow real-time monitoring of brain oxygenation in patients with variable temporal and spatial resolution. Both focal probes are invasive and are single site recording devices, and thus positioning of the probe limits the information obtained. On the other hand, NIRS probes are non-invasive and can assess several regions simultaneously, but can experience contamination of the signal due to extracerebral circulation. Finally, focal PbtO2 probes are considered the most effective bedside method for detection of cerebral ischemia, which is not true for NIRS probes due to lack of standardization between commercial devices and undetermined threshold for ischemia [23].

Material and Methods

81-85 Description of the depth electrode is very limited, a lot of suppositions remains to the reader

R: These are commercially available FDA approved probes and most of the details are protected by patent law. Our analytical characterization pertains to the fact that these are probes designed for monitoring extracellular local field potential (LFPs) while we propose to use them for electrochemical recording of a faradaic currents resulting from redox reactions occurring at the Pt surface. For a sake of clarity, we have added a supplementary figure showing a picture of the multisite electrode so that the reader can have a better understanding of its design and dimension. We have also added some more details according to the manufacture’s specification sheet.

Results

128-129 The Pt and other layers are fabricated by the manufacturer and should be part of the Material and Methods; at least the thickness for the Pt and insulating layers must be known and then investigated in the Morphology and Chemical Analysis section.

R: As mentioned above, the details offered by the manufacturer are quite limited regarding specifications such as Pt thickness and insulating layers.

129 What is the meaning of « albite unevenly »?

R: We have clarified this statement in the manuscript. Our intension was to say that the Pt layer does no completely cover the Aluminium substrate.

Reviewer 3 Report

Ledo et al. deeply characterized the electrochemical properties of commercial neural electrodes, concluding that the electrodes can be used for sensing PbtO2 in vivo. This study can provide valuable database and reference parameters for people who would like to use commercial neural electrodes to do electrochemical sensing in addition to neural recording and stimulation.

  1. I am wondering if the applied potential, -0.6 V, may cause tissue damage during the in-vivo measurement.
  2. I am curious about if the detecting of O2 can suffer from interference from H2O2 which is also an important factor in brain ?
  3. There are writing errors in figure number in the section 3.5 on Page 7.

Author Response

Reviewer 3)

Ledo et al. deeply characterized the electrochemical properties of commercial neural electrodes, concluding that the electrodes can be used for sensing PbtO2 in vivo. This study can provide valuable database and reference parameters for people who would like to use commercial neural electrodes to do electrochemical sensing in addition to neural recording and stimulation.

  1. I am wondering if the applied potential, -0.6 V, may cause tissue damage during the in-vivo measurement.

R: At any electrode surface, the applied potential results in a potential difference between the electrode surface and the surrounding electrolyte solution, which is confined to the electrical double layer and is only a few nm wide. For this particular case, the negative applied potential results in a first layer of positively charges ions at the electrode-solution interface. This is followed by a complex region of ions and at a few nm from the surface bulk solution, where no potential gradient is observed.

The electrochemical reactions and resulting faradaic currents are limited to the electrode surface and do not occur in the bulk solution. Electroactive species, such as oxygen, reach the electrode surface by diffusion, and do not migrate to the electrode surface by a potential gradient due to the high ionic strength of the extracellular medium. Furthermore, amperometric recording does not result in any charge transfer from the electrode to the tissue.

I am curious about if the detecting of O2 can suffer from interference from H2O2 which is also an important factor in brain?

R: This is a pertinent question as H2O2 can be reduced at a Pt surface. With this regards we must consider the concentration dynamic of H2O2 in the brain. For instance, the intracellular concentration of H2O2 is maintained at very low levels (1-100 nM) and extracellular H2O2 is rapidly uptaken into cells and removed by enzymes such as glutathione peroxidase, the peroxiredoxin system and, in vivo, by catalase. Therefore, H2O2 is not expected to act as an interference in the measurement of oxygen in vivo whose concentration is 2 to 3 orders of magnitude higher (around 30-50 µM).

  1. There are writing errors in figure number in the section 3.5 on Page 7.

R: We thank the reviewer for catching this error, which has been corrected.

Reviewer 4 Report

The authors evaluated the electrochemical properties of AuragenTM depth electrode containing 12 recording sites, and the ability to repurpose the electrode for electrochemical detection of O2 for use in a clinical setting. Overall the article is well written and suited for publication in Micromachines, provided that the authors address the minor comments below.

What is the variability of the elemental composition across the 12 recording sites? It is clearly shown in the figures 1F-G, and described in the text but it would be well supported with quantitative data.

Given the variability observed in Figure 1 F-G, is Figure 2 representative of the variability?

Does the variability in the elemental composition of each recording site affect the electrochemical behavior? Is variability in the response observed, despite all electrodes being exposed to the same solution. The authors should include this in the discussion. 

Author Response

Reviewer 4)

Suggestions for Authors

The authors evaluated the electrochemical properties of AuragenTM depth electrode containing 12 recording sites, and the ability to repurpose the electrode for electrochemical detection of O2 for use in a clinical setting. Overall the article is well written and suited for publication in Micromachines, provided that the authors address the minor comments below.

 What is the variability of the elemental composition across the 12 recording sites? It is clearly shown in the figures 1F-G, and described in the text but it would be well supported with quantitative data.

Given the variability observed in Figure 1 F-G, is Figure 2 representative of the variability?

Does the variability in the elemental composition of each recording site affect the electrochemical behavior? Is variability in the response observed, despite all electrodes being exposed to the same solution. The authors should include this in the discussion. 

 R: On average, the bulk recording site is composed of 80-90% Pt, which is shown from the EDS chemical analysis in Fig. 1D and Suppl. F1 and corroborated by the determination of the electrochemical active surface (Pt) and roughness factor (the ratio between electrochemical active surface and the geometrical area, found to be 0.9 +/- 0.1. What Fig. 1F and G represent are not chemical analysis of the buck surface, but rather the chemical composition of the regions in Fig. 1E for spectrum 9 (Al rich region) and 10 (Pt rich region). We have clarified this in the revised manuscript.

So, there is little variability in electroactive area (Pt). However, to correct for this factor, the values of sensitivity (Figure 6A) are corrected and normalized for the calculated electroactive area of each recording site.